

Strong wind occurrence in Poland from the 13th to 16th centuries based on documentary evidence
Rajmund Przybylak[1,4], Andrzej Araźny[1,4], Janusz Filipiak[2], Piotr Oliński[3,4], Przemysław Wyszyński[1,4],
Artur Szwaba[5]
[1] Nicolaus Copernicus University in Toruń, Faculty of Earth Sciences and Spatial Management, Poland
(rp11@umk.pl, andy@umk.pl, przemyslaw.wyszynski@umk.pl)
[2] University of Gdansk, Department of Physical Oceanography and Climate Research, Gdańsk, Poland
(janusz.filipiak@ug.edu.pl)
[3] Nicolaus Copernicus University in Toruń, Faculty of History, Poland (olinskip@umk.pl)
[4] Centre for Climate Change Research, Nicolaus Copernicus University, Toruń, Poland (cccr@umk.pl)
[5] State Water Management, Polish Waters, Toruń, Poland (artur.szwaba@wody.gov.pl)
*Correspondence to*: Przemysław Wyszyński (przemyslaw.wyszynski@umk.com)
**Abstract.** A comprehensive database of strong winds based on documentary evidence was created for
Poland until AD 1600. Three types of documentary sources were used: handwritten and unpublished,
published, and "secondary" literature. The database contains detailed information about the
occurrence of strong winds (the location/region, time, duration and indexation for intensity, extent
and character of damage), as well as the exact textual content of the original weather note, the name
of the source, and an evaluation of the source's quality. Five categories of strong winds were delimited:
1 – fresh and strong breeze (Beaufort scale 5–7), 2 – gale (8–9), 3 – storm (10–12), 4 – squall (i.e., gusty
wind during a thunderstorm), and 5 – tornadoes. The intensity, extent, and character of damage were
estimated based on the proposition given by Brázdil et al. (2004), which we slightly modified to include
the Baltic Sea and its influence on coastal parts. In the database, 137 thus-defined strong winds were
identified. A reliable estimate of some characteristics of the occurrence of strong winds in Poland
seems possible from the mid-15th century onwards. The highest number of strong winds occurred in
the second half of the 16th century, with a maximum in the 1570s. For each season, the greatest
number of strong winds was found for the Baltic Coast and Pomerania region, and then for Silesia and
Lesser Poland. Storms and gales were most common during the cold half-year (mainly in March,
November, and December).
Keywords: historical climatology, documentary evidence, Poland, strong winds, database.

1.  Introduction
Strong winds are among the most significant natural disasters, causing great damage in the entire
world and loss of human and animal lives. In Poland, for example, according to Lorenc (2012), they are
second the most dangerous natural phenomenon after floods. Results presented recently by the
Statista Research Department (Apr 29 2024, https://www.statista.com/statistics/1269886/most-
common-natural-disasters-in-europe) confirm that the same situation is observed in Europe. From
2001 to 2020, floods were most frequent (41%), followed by strong winds (27%). According to
MunichRe (2011, 2020) estimates, ~60% of all insured losses during 2000–18 were due to extreme
meteorological events, primarily extreme winds. Cusack (2023, see Figs 2 and 7) estimated the annual
windstorm losses in Europe (12 countries) from 1950 to 2022. The results showed that yearly losses



usually (~80%) oscillated between 1 and 5 billion euros. Although no long-term trend is seen in the
study period, the greatest losses were observed in the 1980s.
Good and reliable knowledge about extreme winds is essential for many economic sectors,
e.g., the design and construction of large and high buildings or the wind power sector (Outten and
Sokolowski 2021). However, our knowledge about the different characteristics and impacts of extreme
winds is still based mainly on results gathered for the instrumental period (more or less for the last
100–150 years), and most often only for the last few decades. Therefore, such knowledge is still
insufficient and needs improvements, which can be done by using a more extended series of data
coming from the early- and pre-instrumental period. Many such data are still undiscovered or
discovered but not digitised and exist only on paper stored in numerous archives worldwide (Hawkins
et al. 2003). That is why data rescue activity is vital (for details, see e.g. Brönnimann et al. 2019;
Lundstad et al. 2023). It can help improve our understanding of historical climate variations, including
strong winds. For example, Hawkins et al. (2019) demonstrated that the severe windstorm that
occurred in February 1903 in England and Wales (reconstructed by them using documentary evidence)
was characterised in some places by stronger winds than were observed in the modern period (1950–
2015). They thus suggest that an estimate of risk from severe windstorms based on contemporary data
may need to be revised. A longer perspective on changes in intensity and impacts of extreme winds
can also be beneficial for their future simulations and also for the more reliable assessment of the risks
connected with them for societies. However, significant spatial-temporal changes of strong wind
occurrences and also their rarity and often local character significantly hinder the proper identification
of the mechanisms responsible for their changes and risks.
According to Donat et al. (2011), in future climate simulations (investigated using multi-model
simulations from global [GCM] and regional [RCM] climate models), enhanced extreme wind speeds
were found over northern parts of Central and Western Europe in most simulations and in the
ensemble mean (up to 5%). Consequently, they forecast that the potential losses will be higher in these
regions, particularly in Central Europe. In turn, in Southern Europe, according to them, an expected
decrease in extreme wind speeds will result in a reduction in loss potential. More recent work (Outten
and Sokolowski 2021) partly confirms the above findings. Using a 15-member ensemble of high-
resolution Euro-CORDEX simulations (~12 km), they found increases in the return period, i.e. more
frequent extreme episodes projected for Northern, Central and Southern Europe throughout the 21st
century. At the same time, they underlined, however, that the assessments of future extreme wind
changes remain fraught with uncertainty.
As seen from all the presented scenarios, strong winds in Poland will be more common. Thus,
associated economic and societal consequences may also be more significant than at present.
Therefore, the investigation of all the characteristics of strong winds and their impacts should be



intensified in Poland. Although we have quite a large number of works describing strong winds in
Poland in the contemporary period using instrumental measurements (e.g. Stopa-Boryczka 1989;
Paszyński and Niedźwiedź 1991; Krawczyk 1994; Adamczyk 1996; Lorenc 1996, 2012; Araźny et al.
2007; Tarnowska 2011; Ustrnul et al. 2014; Chojnacka-Ożga and Ożga 2018; Wibig 2021 and references
therein) they all cover some periods since 1950s and are written mainly in Polish and therefore are
unknown to the international scientific community. Only in a few works (e.g., Bartnicki 1930; Gumiński
1952; Piasecki 1952) is analysis of winds (including strong winds) available for an instrumental period
before 1950, starting from the late 19th century. On the other hand, there is a complete lack of such
works for historical periods, i.e. before 1800, using documentary evidence and existing visual regular
observations of winds. For example, the latter kind of observation with a quantitative estimation of
the force of the wind using a seven-degree scale (0–6) exists for Wrocław (Orig. Breslau, source:
newspaper *Oekonomische Nachrichten der Patriotischen Gesellschaft in Schlesien*) and a five-degree
scale (0–4) for Żagań (Orig. Saganenses, source: *Ephemerides Societatis Meteorologicae Palatinae*,
*1783–1795*, see also Przybylak et al. 2014 and Pappert et al. 2021) for the periods 1773–81 and 1781–
92, respectively. In addition, similar detailed observations of wind reporting quantitative force of wind
also exist for Gdańsk, where the first regular meteorological observations started in 1655 and were
carried out by Bűthner, a professor of mathematics, who used to note daily observations of various
weather phenomena including the occurrence of strong winds. Unfortunately, his manuscript was lost,
probably irretrievably. It is possible to analyse only data for some selected years within his
observations covering the years 1655–1701. The 18th century (the Enlightenment) brought to the
coast a boom in interest in observing weather conditions and their impact on the economy. Its climax
was the beginning of regular instrumental meteorological measurements in Gdańsk by Hanov on the
1st of January 1739. Wind observations were made using a nine-degree scale (0–8) in 1739–72 (see
Table 5.1 in Przybylak 2010).

104        Knowledge about strong wind occurrences in Europe in the pre-instrumental period is also very

limited, although significantly better than in other parts of the world. For example, for the period under
study (13th–16th centuries), most of the works containing the most detailed climatic analyses
regarding strong winds are available mainly for the Czech Lands (e.g., Brázdil and Dobrovolný 2000,
2001; Dobrovolný and Brázdil 2003; Brázdil et al. 2004 and references therein). The last item is
particularly valuable for its very detailed analysis of different aspects related to strong winds for the
entire last millennium based on documentary evidence. However, even in this publication, some
limited information about strong winds for the pre-1500 period is available, directly caused by the
small number of existing historical sources. For central Europe, information about strong winds is also
contained in monographs analysing different kinds of extremes (Pfister 1999; Glaser 2001, 2013). For
the Low Countries (the coastal areas of the southwest Netherlands and Flanders), a valuable paper is



available presenting storminess changes in the period 1390–1725 (de Kraker 2013 and references
therein). It is essential to add that the author also graded storm events using an eight-degree scale.
Similar work, as mentioned for the Low Countries, also exists for the North Sea, the British Isles, and
Northwest Europe (Lamb 1991 and references therein). For a smaller area in this European region
(Thames estuary), there are also works published by Galloway and Potts (2007) and Galloway (2009).
Finally, we should also mention the work of Orme (2015), which analysed late-Holocene storminess in
Europe using various proxies.
This short review shows that our knowledge about the occurrence of strong winds in Europe,
although better than in other parts of the world, is very limited. That is why there is an urgent need to
improve and widen this knowledge. The main aim of the present paper is to partly fill this gap by
presenting an analysis of strong wind occurrences in Poland for the period from the late 13th century
(the first record of strong wind in Poland found in historical sources) to the end of the 16th century.

**2.  Area, data and methods**
129          2.1. Area

The analysis of strong winds in the studied period is conducted for the area of Poland within
contemporary boundaries. Poland is a Central European country stretching from the Baltic Sea in the
north and the Sudetes and Carpathian Mountains in the south (Fig. 1). To more precisely estimate the
spatial changes in strong winds occurrence, the analysis was also made for six historical-geographical
regions: Baltic Coast and Pomerania, Masuria and Podlasie, Greater Poland, Masovia, Silesia, and
Lesser Poland (Fig. 1). Data from contemporary period representing all the mentioned regions were
collected for 12 meteorological stations – two for each region.

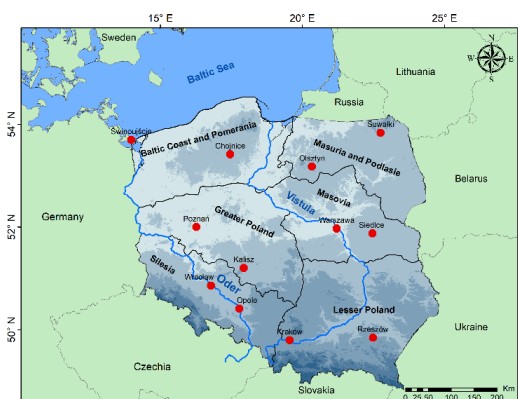


Fig. 1. Geographical location of Poland, main historical-geographical regions and contemporary meteorological
stations (red dots) (after Ghazi et al. 2024, modified)





140       At present (1966–2018), the average annual wind speed calculated based on 41 stations was

3.6 ms$^{-1}$ (Wibig 2021). Excluding mountainous areas, for which we have no historical data, the largest
average wind speeds in Poland are noted in the coastal part of the Baltic Sea (4–5 ms$^{-1}$) and a little
smaller in the central part, e.g. in Warszawa (Eng. Warsaw) 4.0 ms$^{-1}$. The weakest winds in Poland are
noted in the foothills of the Sudeten and Carpathian Mountains (less than 3 ms$^{-1}$). The spatial
distribution of extreme winds is slightly different compared to the average ones. The greatest extreme
winds occur in the SW part of Poland and then in the coastal part of the Baltic Sea (Wibig 2021).

147       2.2. Sources and data

The following three types of documentary sources were used: handwritten and unpublished,
published, and "secondary" literature (e.g., articles, monographs) to search for weather notes
describing the occurrence of strong winds in Poland. The number of used historical sources in the study
period correlates strongly with their availability, which is the greatest in the less-distant centuries. For
example, for the 16th century, we used 85 sources, while for the 13th and 14th centuries, we used 1
and 27 sources, respectively. For every event of strong wind occurrence in Poland, a detailed reference
to the source(s) is given; see https://doi.org/10.18150/W6PMBQ.

155       A quality assessment of each used source was conducted by historians using the method called

"source criticism" in the historical sciences. This method allows for examining the authenticity of
specific human activities (historical source) and reading their meaning (historical event) in the light of
the causes and conditions of their creation in the historical process. Three quality categories were
distinguished: 1 – weak, 2 – moderate, and 3 – high, to choose an appropriate source and weather
note(s) describing strong winds. The following rules were used to stratify sources according to their
quality; 1 – weak, if the information was derived from secondary literature rather than the original
source; 2 – moderate, if the information was written centuries after the strong wind occurrence; and
3 – high, if the information was written in a source in the same period that the strong wind event
occurred and provides precise information. For the analysis, we used mainly the last category of
sources, which provided the best valuable data.

167       2.2.1.   Database: historical period

For the first time, a comprehensive database of strong winds related to the period before the 19$^{th}$
century is prepared for Poland by a team of climatologists and historians based on all documentary
evidence. The database at the present stage is finished until the end of the 16$^{th}$ century (see
https://doi.org/10.18150/W6PMBQ). It contains detailed information about the occurrence of strong
winds (the location/region, time, duration, and indexation for intensity, extent, and character of





damage), as well as the exact textual content of the original weather note, the name of the source,
and an evaluation of the source's quality. The information is not complete for every case of strong
wind. Sometimes, only general information is available about the strong winds in Poland. In such a
case, there is no information about the place or region; therefore, we introduced an additional region
category called "Poland". Information available for the time of the strong wind occurrences is also
varied. For some, we only have information about the year; for others, we only have information about
both season and year. However, in most cases, we have information about year, month and day(s)
(and often the start and end of the phenomenon). For this reason, it is essential to remember that
presented statistics, e.g. frequency of occurrence of strong winds in months, seasons and years, are
based on different numbers of cases. In Table 1 we showed some selected examples of entries to the
database presenting varying degrees of detail.






Table 1. Examples of entries in the strong wind database. Explanation of abbreviations (A, B, C, D) below the table. Explanations of numbers and abbreviations in columns A, B, C, and D are provided in the database.

| Region | Place | Date of occurrence | Description: original | Description: translation | Source | A | B | C | D |
|---|---|---|---|---|---|---|---|---|---|
| A complete record, including details of the exact date and place of occurrence of the phenomenon and associated damage | | | | | | | | | |
| Baltic Coast and Pomerania | Darłowo, Western Pomerania | 11–13 Jan 1558 | 1558 Am Dienstag nach Maria Lichtmess (8. Februar) richtete der Wind in Stralsund grossen Schaden an den Brücken der Stadt an, die alle bis auf eine entzweibrachen. Auch Schiffe und Botte wurden zerstrümmert. Es war Nordwestwind, und zwar in der Nacht, sonst wären auch wohl Menschen ertunken. Auch auf dem Lande wurde den Häusern und Obstbäumen viel Schaden zugefügt. [...] Auch in Rügenwalde richteten zwei grosse Stürme, die vom 11. bis 13. Januar und am 8. Februar wüteten, viele Verheerungen an. Das Wasser soll ellenhoch an der Stadtmauer gestanden heben, in die Speicher bei der Wipper gedrungen sein und auf de Münde 18 Wohngebäude zerstört haben, deren Einwohner sich nur dadurch retteten, dass sie auf Bäume kletterten. | 1558 On the Tuesday after Candlemas (February 8th), the wind in Stralsund caused great damage to the city's bridges, all except one broke in two. Ships and boats were also wrecked. It was a north-westerly wind, and at night, otherwise people would probably have drowned. Much damage was also done to houses and fruit trees in the countryside. [...] Two big storms that raged from January 11th to 13th and February 8th also caused a lot of devastation in Rügenwalde. The water is said to have risen several metres above the city walls, infiltrated the granaries near the Wipper and destroyed 18 residential buildings on the estuary, whose residents only saved themselves by climbing trees. | Besch R., Strenge Winter in alter Zeit. Witterungsgeschichtliches aus Pommern, Unser Pommerland, Jg. 8 (1923), H. 1, p. 13. | 1 | 3 | 2 | DB DS DU DV DF |
| Inaccurate record – only fragmentary information on date of occurrence and associated damage | | | | | | | | | |
| Silesia | Nysa | autumn 1578 | Ein heftiger Wind beschaedigte im Spaetherbst den obern Theil des Rathhauses, das schon durch fruehre Stuerme und Gewitter gelitten. | In late autumn, a strong wind damaged the upper part of the town hall, which had already suffered from early storms and thunderstorms. | Minsberg F., Geschichtliche Darstellung der merkwuerdigsten Ereignisse in der Fuerstenthums Stadt Neisse, Neisse | 1 | 1 | | DS |
| General record - very fragmented information on the phenomenon | | | | | | | | | |
| Baltic Coast and Pomerania | Koszalin | 1531 | Anno 1531 — War ein grosser Sturm Wind, dass der Wind den grossen Wetter Hahn vom Kirchen Thurm abwarf und haben einige dieses vor eine Vorbedeutung gehalten, dass die Catholischen sollten untergehen, so auch kurtze Zeit darauf richtig erfolget, nemlich anno 1534, da das Pabsthum im gantzen Lande abgethan worden. | Anno 1531 – There was a great storm wind that threw the great clock from the church tower, and some thought that this was an omen that the Catholics would perish, which happened a short time later, namely in 1534, when the papacy was abolished in the entire country. | Wendtland J.D., Eine Sammlung unterschiedlicher die Historia der Stadt Cöslin betrffende Sachen | 3 | 3 | 2 | DB |





The entire database contains 137 records documenting the occurrence of strong winds in
Poland. The first weather note reporting the presence of strong wind was found for the year 1283. The
number of weather notes (162) is not identical to the number of strong wind cases because sometimes
we have more than one weather note describing the same case of strong wind. The statistics of
weather notes documenting the occurrence of strong winds are presented in Fig. 2. The greatest
number of weather notes we found for the 16th (85 cases) and 15th (52) centuries and less for the 13th
century (1). Most of them we found for autumn (63) and winter (35). The majority of weather notes
exist in the documentary evidence for two regions, Baltic Coast and Pomerania (93) and Silesia (39),
and least for the Masuria-Podlasie and Masovia regions, with 1 and 2, respectively (Fig. 2).

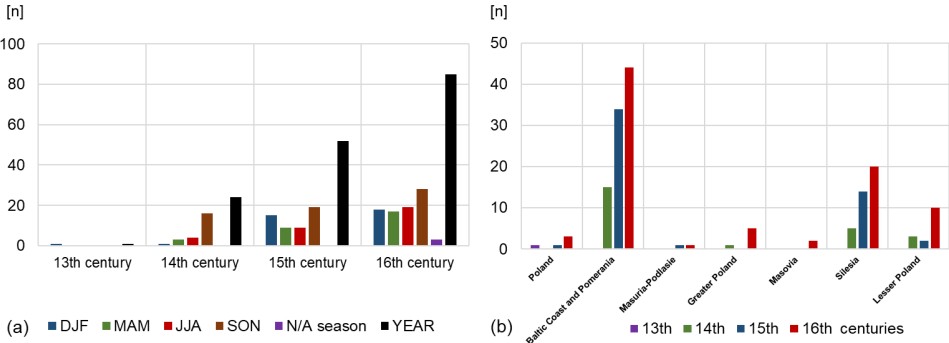

Fig. 2. Number (n) of weather notes for Poland for seasons (a) and regions (b), 1281–1600
There is strong coherence between the number of available sources and the number of
weather notes (usually the number of weather notes is greater than the number of sources) on the
one hand and the number of occurrences of strong winds on the other. This pattern is similar to what
we observed for flood records (Ghazi et al. 2023a, b, 2024, 2025) and drought records (Przybylak et al.

21   2020).

2.2.2.   Database: contemporary period
Sub-daily data (every three hours) of wind speed for the period 1993–2022, for which most
homogeneous and complete series of wind values for Poland are available, were gathered for 12
meteorological stations (see Table 2, Fig. 1). As mentioned earlier, they also represent all six
distinguished historical-geographical regions in the country. The wind data were downloaded from the
website of the Institute of Meteorology and Water Management - National Research Institute (IMGW-
PIB) (https://danepubliczne.imgw.pl/). The two data types were collected as average wind speed every
three hours and the highest gust of wind for 3-hour intervals.



Table 2. Geographical location of meteorological stations used in the work

| No. | Name of meteorological station | H (m a.s.l.) | φ (N) | λ (E) |
|---|---|---|---|---|
| 1. | Świnoujście | 6 | 53°55' | 14°14' |
| 2. | Chojnice | 164 | 53°43' | 17°33' |
| 3. | Olsztyn | 133 | 53°46' | 20°25' |
| 4. | Suwałki | 184 | 54°08' | 22°57' |
| 5. | Poznań | 83 | 52°25' | 16°51' |
| 6. | Kalisz | 138 | 51°47' | 18°05' |
| 7. | Warszawa | 106 | 52°10' | 20°58' |
| 8. | Siedlce | 152 | 52°11' | 22°15' |
| 9. | Wrocław | 120 | 51°06' | 16°53' |
| 10. | Opole | 165 | 50°38' | 17°58' |
| 11. | Kraków | 237 | 50°05' | 19°48' |
| 12. | Rzeszów | 200 | 50°06' | 22°03' |


2.3. Methods


2.3.1.    Historical period

In our analysis, we distinguished four categories of strong winds, which are consistent with the
proposition used by Brázdil et al. (2004, their types T1, T3–T5 in Table 6.1) for Czech Lands. In addition,
we added one more category (tornadoes), treated separately:

a) Fresh and strong breeze (force according to Beaufort scale [BS] 5–7),

b) Gale (BS 8–9),

c) Storm (BS 10 and more),

43        d) Squall (i.e., gusty wind during a thunderstorm),

e) Tornado.

Using the final version of the database, each case of strong wind was analysed in detail. In the

first stage, an indexation of its intensity was done by the author of a particular record contributed to
the database. In the second stage, the proposed categories of wind intensities by individual
contributors were discussed and finally accepted by the whole team of authors of this paper. To
investigate the damage caused by strong winds, three categories of extent of damage were utilised
(consistent with types E0–E2 in Brázdil et al. 2004, Table 6.2, modified):

a) E0 – no information about damage,

b) E1 – small damage, damage of lesser extent,

c) E2 – large damage, areally extensive damage.

The last category was slightly modified by us to include damages on the sea (destruction or sinking of
ships) and losses caused by storm floods – reflecting the coastal location of Poland. Finally, if weather





notes allowed, we also estimated the damaged character, again using the proposition of Brázdil et al.
(2004, Table 3).

Similarly as in the case of the extent of damage, we added two more categories to the list of

types of the character of damage presented for the Czech Lands. The proposed new categories
precisely describe the influence of the Baltic Sea on coastal parts. The "N/A" designation was used
when the weather notes did not contain information about the character of damage. Thus, the
following nine categories were distinguished to characterise damages and losses:

DO – casualties (lost lives),

DL – wind damage in forests,

DP – minor damage to buildings,

DB – considerable damage or destruction of buildings,

DS – uprooted fruit trees, damage to hop gardens and vineyards,

DU – damage to field crops, gardens and orchard harvests,

DV – considerable damage/destruction to vessels (including sunk),
newly added

DJ – other damage (e.g., upturned carriages, vessels, injured persons, minor damage to
property),

DF – considerable damage/destruction by storm flood or inland flood, newly added,

N/A – information not available.


2.3.2.    Contemporary period

Based on gathered sub-daily wind data, the following statistics were calculated:
1. Average daily, monthly, seasonal, and annual wind speed,
2. Highest gust of wind for every day, month, season, and year,
3. Frequency of gust winds > 17.0 ms$^{-1}$ calculated for every month and year for each station,
4. Frequency of gust winds in the following speed intervals:

a)  8.0–17.1 ms$^{-1}$ (BS 5-7) – quite strong, strong, and very strong wind,

b)  17.2–24.4 ms$^{-1}$ (BS 8-9) – gale,

c)  >24.4 ms$^{-1}$ (BS 10-12) – very strong and violent storms, and hurricanes.

These three categories of strong winds listed in point 4 above, but particularly the categories
described in points b and c, were used for comparison with the historical data. The analysis, however,
is limited mainly to estimating differences and similarities in the study area's annual course and spatial
distribution. We need to add that a reliable comparison of absolute values of the frequency of strong





wind occurrences in both periods is impossible due to the undetectability of probably a significant
number of events in the study's historical period, particularly those classified to the first category.
For many years the IMGW-PIB did not maintain a separate database on squalls, land- and
waterspouts and tornadoes, and the damage caused by the occurrence of such extreme events. In fact,
this was a typical measure for many European national meteorological services – the duty to monitor
damage caused by the aforementioned phenomena belonged to other state services. However, in
2015, the WMO decided to standardize weather, water, and climate hazard information to allow more
sophisticated analyses of data on the occurrence of such phenomena and the losses and damage
associated with them. Then, many meteorological services, including IMGW-PIB, started to catalogue
extreme weather, climate, and water phenomena and develop a dedicated database, but the material
collected so far is too sparse for us to make a comprehensive analysis.

3.  Results
3.1. Historical

According to Przybylak et al. (2023), in Poland, the Medieval Warm Period (MWP, recently also called
the "Medieval Climate Anomaly", MCA) started in the late 12th century and finished between the mid-
14th and mid-15th centuries. Then, until the mid-16th century, the Transitional Period (TP) was
distinguished (Niedźwiedź et al. 2015), followed by the Little Ice Age (LIA). Thus, our study period
covers a large part of the MWP, the entire TP and the early decades of the LIA. As results from analysis
conducted by Przybylak et al. (2023) and also from the database accompanied by this paper, the
available number of weather notes allow for a reliable estimate of the occurrence of strong winds,
mainly from the 1450s onward. Two maxima of greatest storminess can be distinguished in this time:
1451–1520 and 1551–1600 (Fig. 3). The latter, however, reveals a greater number of occurrences of
strong winds, in particular in two decades: 1571–80 (14 cases) and 1581–90 (12). In the pre-1450
period, we should underline the clear maximum of strong winds noted in the first half of the 14th
century, followed by a less stormy period until the 1450s.





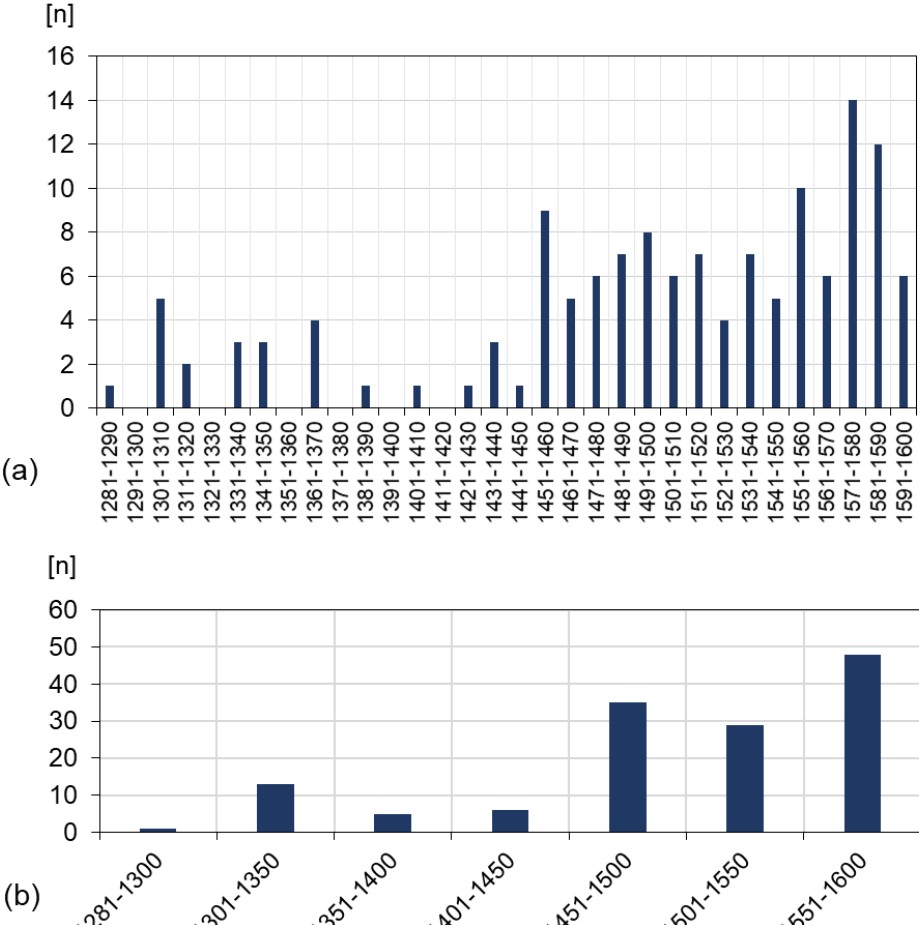

(a)

(b)


Fig. 3. Decadal (a) and 50-year (b) number (n) of occurrence of all categories of strong winds in Poland, 1281–1600


Out of the six regions analysed, strong winds were noted most often in the Baltic Coast and

Pomerania region (74 cases), and then in the Silesia region (34) (Fig. 4a, b). Quite often, they were
registered also in the Lesser Poland region (15). In other regions their occurrence was sporadic, ranging
from 1 to 6 cases (Fig. 4b). The annual cycle of strong winds occurrence can be estimated based on
regions for which enough information exists. Strong winds were noted most often in autumn and in
winter for Silesia and the Baltic Coast and Pomerania regions, and in summer and fall for the Lesser
Poland region (Fig. 4b). August (13 cases) and March (12) were most abundant in strong wind
occurrences in Poland, but the stormiest period of the year was from October to March, with at least
ten cases in each month except February (Fig. 4a).



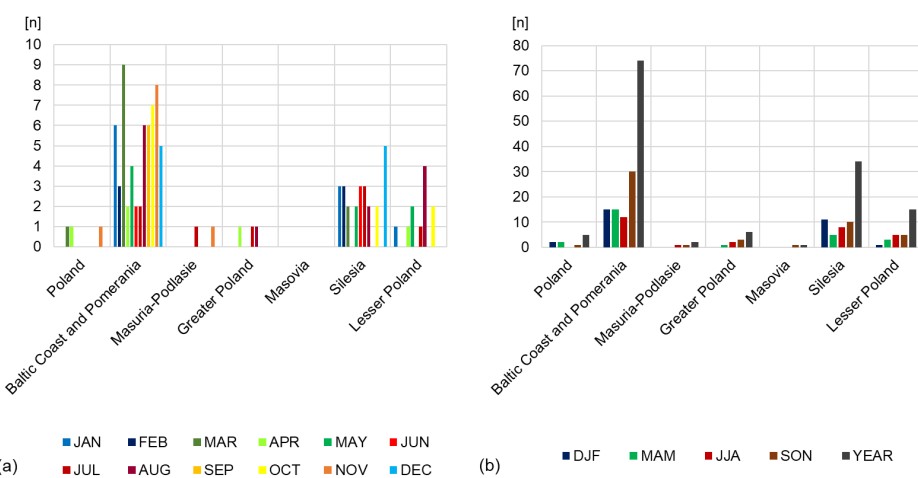


Fig. 4. Monthly (a) and seasonal (b) number (n) of all categories of strong winds according to regions, 1281–
1600. Explanation: Please note that the total number of strong winds in seasons/years presented in Fig. b is greater than calculated based only on monthly statistics because, in some weather notes, there is information only about the season or even the year of the strong wind occurrence (see also text in 2.2.1. and Table 1).

In the entire study period (1281–1600) (Fig. 5), and also in two subperiods (1281–1500 and 1501–1600) (Fig. S1), the most frequent were storms and gales and the least frequent were fresh and strong breezes and, in particular, tornadoes (only one case). The storms and gales were most common in spring (mainly in March), autumn (particularly in November), and winter (mainly in December) (Fig. 5). It is worth noting a big change between the studied two sub-periods in August. In the period 1281–1500, in this month only storms were registered, whereas in the 16th century, all categories of strong winds were recorded (except tornadoes) (Fig. S1).

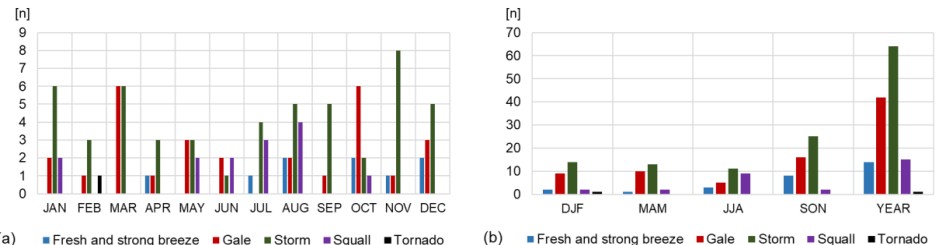

Fig. 5. Monthly (a) and seasonal (b) number (n) of occurrence of different types of strong winds in Poland in the period 1281–1600

The spatial distribution of occurrence of strong winds (except tornadoes) is presented in Fig. 6 for the entire study period and in Fig. S2 for two sub-periods. The most significant differences in spatial distribution between the two sub-periods are seen for two categories of strong winds, i.e. fresh and



strong breezes and (particularly) for squalls (Fig. S2). In the latter case, the squalls were noted in the
historical sources in the period 1281–1500 only for three regions: Silesia, the Baltic Coast and
Pomerania, and Lesser Poland. In the 16th century, they were also found in Greater Poland. On
average, for the entire study period, all categories of strong winds were most frequent in the Baltic
Coast and Pomerania region (about 46–69%), except the squall category, which was noted most often
in Silesia (40%). The region with the second greatest amount of strong winds (except squalls) was
Silesia (14–27%) (Fig. 6).

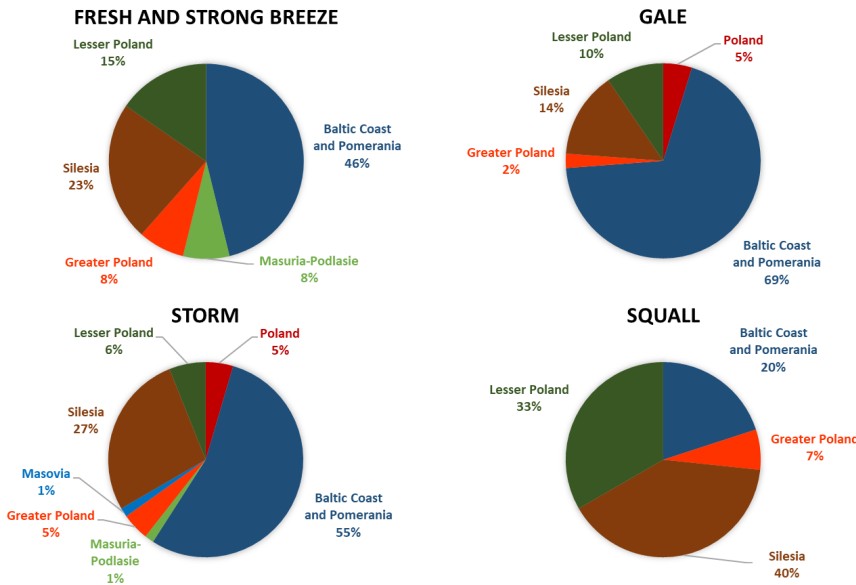


Fig. 6. Relative frequencies (%) of occurrence of different types of strong winds (tornadoes excluded) in the
studied regions of Poland, 1281–1600

The first category of strong winds (category 1) we proposed does not cause damage. The wind

is chaotic in its flow, causing the moving of large tree branches and whole trees; it makes the use of
umbrellas and walking against the wind difficult and causes snowstorms and blizzards during snowfall.
According to Lorenc (2012), a gust wind speed above 17 ms$^{-1}$ in climate conditions in Poland creates a
threat to the population, economy, and environment. It means that all gales and storm winds
(categories 2 and 3) classified by us based on documentary evidence have the potential to cause
various kinds of damage, which we listed in the method section. However, when chroniclers mention
these winds in historical materials, they rarely mention the devastating effects of their impact on the
environment or humans. The lack of information about the destruction probably relates to the
immediate area of their residence and does not rule out possible destruction in other locations. In the
case of squall winds, due to their locally limited character and when there is no information about





damage, we cannot say the same, as in the case of gales and storms that are the effect of
macrocirculation. The available weather notes describing the occurrence of strong winds allow us to
classify damage according to its scale, small or large (see Fig. 7 and Fig. S3).

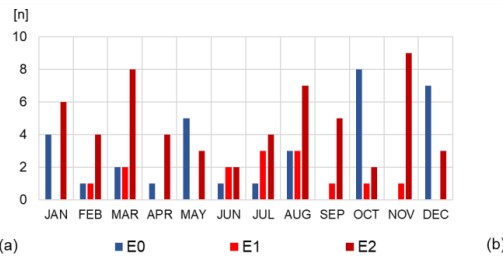

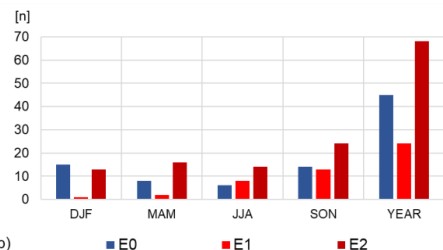


Fig. 7. Monthly (a) and seasonal (b) number (n) of occurrence of strong winds in Poland, according to the extent
of damage, 1281–1600
It is interesting to note that, for the study period, category E2 of damage is most frequent,
then category E0, i.e. no information about damage. Category E2 prevails in all seasons (excluding
winter) and in most of the months, except December, October, June and May (Fig. 7). It is also worth
noting that the frequency of category E0 in relation to E2 decreases from medieval times to the 16th
century (see Fig. S3). This means that the 16th-century chroniclers were inclined to describe weather
events, in this case, strong winds, and their effects more precisely than their predecessors.
In the entire dataset, the largest share of strong winds causing damage (categories E1 and E2)
and in which there is no information on damage (category E0) was found, in line with expectations, for
the Baltic Coast and Pomerania (39–67%) and Silesia (16–35%), for which the most abundant datasets
exist (Fig. 8). A similar situation is noted for both sub-periods with one only exception being the
category E1 in the period 1281–1500 (Fig. S4). This category was a little more frequently reported for
Lesser Poland (29%) and Silesia (29%) than the Baltic Coast and Pomerania (28%) region.

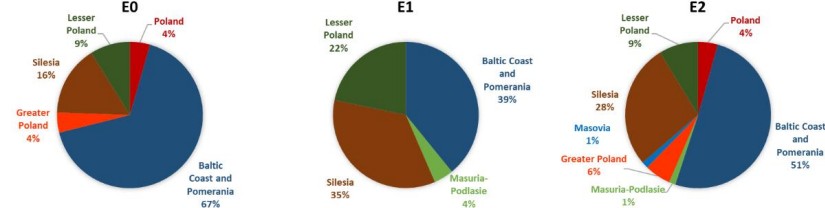



Fig. 8. Relative frequencies (%) of occurrence of strong winds in studied regions of Poland for different types of
damage, 1281–1600. For explanations of abbreviations, see Methods section.
The character of damage caused by the strong winds is shown in Fig. 9 for the entire study period and
in Fig. S5 for the two analysed sub-periods. The two figures are roughly similar.



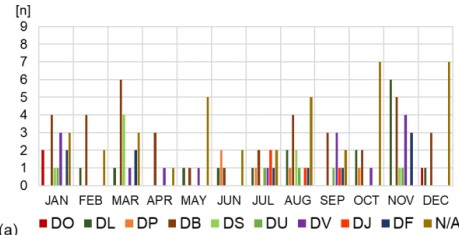

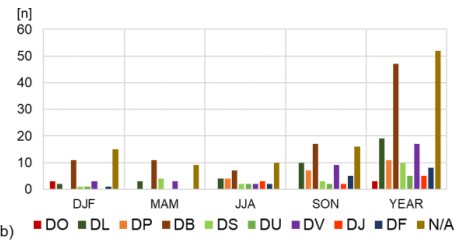

(a) ■ DO ■ DL ■ DP ■ DB ■ DS ■ DU ■ DV ■ DJ ■ DF ■ N/A          (b) ■ DO ■ DL ■ DP ■ DB ■ DS ■ DU ■ DV ■ DJ ■ DF ■ N/A


Fig. 9. Monthly (a), seasonal and annual (b) number (n) of strong winds in Poland for which
information about the character of damage exists (DO, DL, DP, DB, DS., DU, DV, DJ and DF) or does not exist
(N/A), 1281–1600. For explanations of abbreviations, see the Methods section.
On average, 28.7% of weather notes did not contain information about the damage caused by the
strong winds (Fig. 9). Fewer such cases occurred in the 16th century (24.5%) than in medieval times
(34.2%) (Fig. S5). Out of all distinguished damage categories, DB was the most frequent (26.0%), with
a maximum in the 16th century (28.4%). The following most frequent categories of damage noted in
the study period were DL (10.5%) and DV (9.4%), while the least frequent was DO (1.7%) (Fig. 9). About
60% of all damages mentioned in the weather notes were found for the cold half-year, but particularly
for autumn (39.2%). The smallest number of weather notes about damages was noted for spring
(16.6%). In the annual course, damages caused by strong winds in Poland occurred most often in
November (20 cases, 14.0%), March and August (17, 11.9%), with the smallest frequency in April (5,
3.5%) (Fig. 9a).
The spatial distribution of distinguished categories of damage made by strong winds in Poland
in analysed regions is shown in Fig. 10 for 1281–1600 and in Fig. S6 for two sub-periods: 1281–1500
and 1501–1600. The analysis reveals that the most frequently noted categories of damage in Poland
(DB, DL) were also present in all analysed regions, with a maximum in Silesia and Baltic Coast and
Pomerania regions. For these two regions, the number of information items about other categories of
damages caused by strong winds was also the highest, except for the DJ category, which was noted
most often for the Lesser Poland region (Fig. 10). The greatest number of weather notes describing





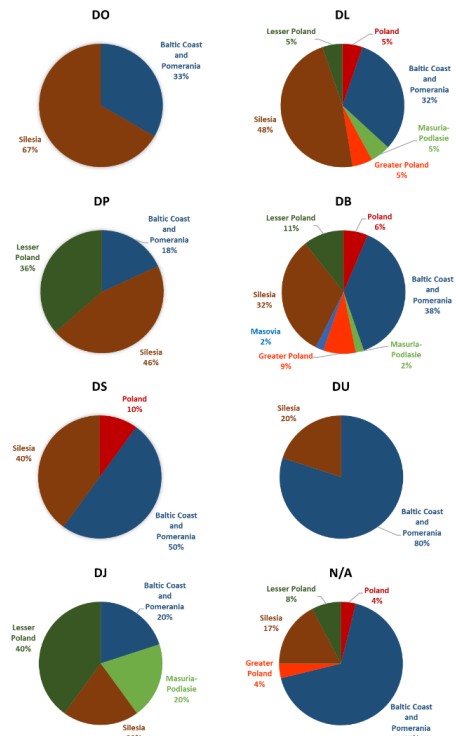


Fig. 10. The relative frequencies (%) of damage categories estimated for particular regions in Poland in the period 1281–1600. Explanations: Two categories (DV and DF) are not shown because they can occur only in the Baltic Coast and Pomerania region. For explanations of abbreviations, see the Methods section.

strong winds not informing about damages was found for the Baltic Coast and Pomerania region (67%) and then for the Silesia region (17%). The spatial distribution of damages caused by strong winds presented for the entire study period was more similar to that noted in the 16th century than that in the medieval period (cf. Figs 10 and S6).

### 3.2. Contemporary period

A short analysis of the present occurrence of strong winds in Poland can be helpful for some comparison purposes with historical results, mainly spatial distribution and run of the frequency of strong winds in the annual cycle. Also, helpful knowledge can be about the values of the greatest speed of strong winds observed presently in Poland.

The annual number of gust winds >17 ms$^{-1}$ in Poland in 1993–2022, which can potentially cause damage, usually does not exceed 20 cases (Fig. 11), but 30-year average values in all stations are <10, except Rzeszów. The probability of occurrence of that kind of strong wind was the smallest in Masovia and Silesia (fewer than 16 cases). There are significant fluctuations in the annual number of gust winds





from year to year, sometimes exceeding 10 cases. Also, the number of thus-defined strong winds has
decreased since about 2010, particularly in the Masovia and Silesia regions.

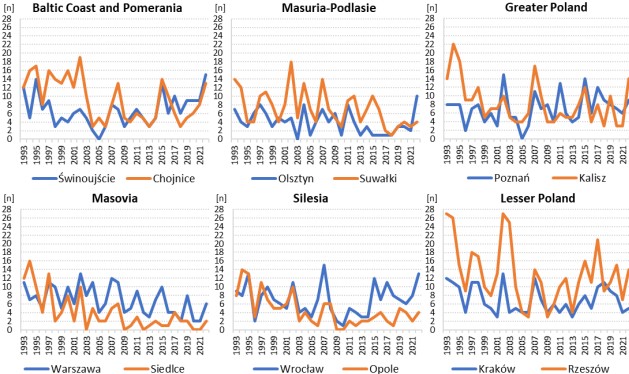


Fig. 11. Year-to-year course of the annual number (n) of gust winds > 17 ms$^{-1}$ in Poland, 1993-2022

In the annual cycle, the average 30-year (1993–2022) monthly number of strong winds >17

ms$^{-1}$ oscillated from below 0.4 in the warm half-year to more than 0.8–1.0 in Jan–Mar (Fig. 12). The
first half of the cold season (Oct–Dec) has significantly fewer gust winds, rarely exceeding 0.8 cases in
one month, except December in some stations.

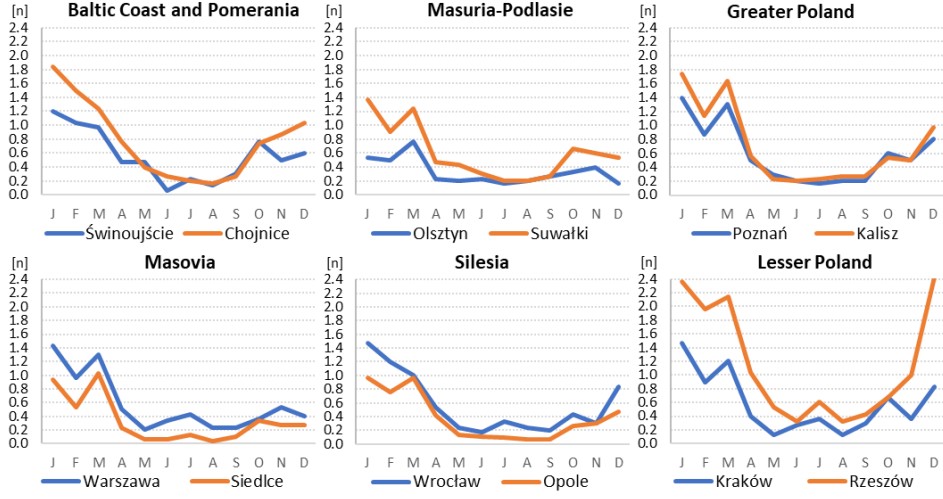


Fig. 12. Annual course based on the average monthly number (n) of gust winds >17 ms$^{-1}$ in Poland, 1993–
2022

In the studied period, the highest gust wind speed in the majority of analysed stations

exceeded 30 ms$^{-1}$. It occurred most often in winter or summer months (Fig. 13). On the other hand,
the lowest, oscillating between 16 and 22 ms$^{-1}$, were measured only in the period from May to

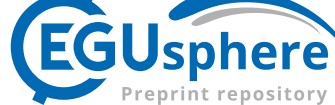

September. The highest strong wind speed (35 ms⁻¹) in Poland was measured in Wrocław (SW Poland)
on July 23, 2017, while the lowest (18 ms⁻¹) was recorded in a few places (Świnoujście – June 18, 2012
and June 28, 2014; Siedlce – May 4, 1996 and May 3, 1997; Opole – August 28, 1994 and August 19,
2022). Gust winds >17 ms⁻¹ stratified into gales (8–9 BS) and storms (10 BS and more) do not show any
important changes in the annual courses compared to annual courses based on all cases of strong
winds >17 ms⁻¹ (see Fig. 12 and Fig. S7). The main reason for this is the rare occurrence of storms in
Poland. In the study period, storms were not observed in most of the years, and the maximum of them
in one year reached three cases (not shown).

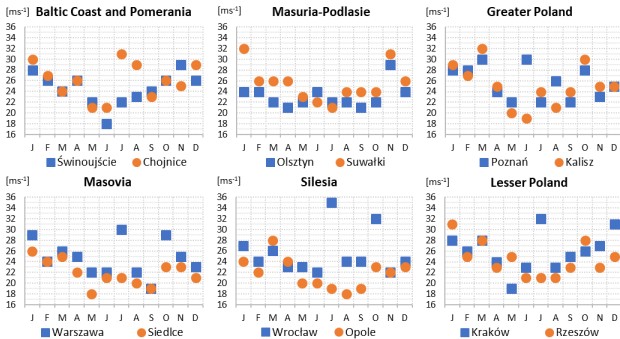

Fig. 13. Maximum monthly wind gust speeds (ms⁻¹) at selected stations in Poland in 1993–2022

4. Discussion
Five categories of strong winds were distinguished in our database
(https://doi.org/10.18150/W6PMBQ, see also Methods) and analysed in the present paper. The first
three categories of strong winds (fresh and strong breezes, gales, and storms) in central Europe are
connected with vigorous moving cyclones born near Iceland, and in particular with a passing cold front.
Cyclonic activity is the greatest in the cold half-year (October to March), and therefore, these
categories of strong winds dominate in this part of the year. On the other hand, the last two categories
(squalls and tornadoes) are typical for the warm half-year (April to September) and are related to
thunderstorm clouds (Cumulonimbus) developing within the thermal turbulence in the one air mass
(sometimes called "isolated" or "local" thunderstorms) or within the zone of passing cold front
(multiple-cell storms, squall lines, or a supercell). As a result, the duration of these kinds of strong
winds is short, and their spatial coverage is local, in particular in the case of the first category.
Recognising the types of strong winds based on the available descriptions of this weather
element in historical sources is not always easy and unambiguous, due to the scarcity of information.
Another difficulty in analysing changes in the frequency of occurrence of the phenomenon is the



increasing number of sources as we move closer towards the present day. According to Brázdil et al.
(2004), this is the main limiting factor for the climatological analysis investigating changes in the
occurrence of strong wind in historical times. Moreover, in older periods, attention was focused more
on describing only extremely strong winds (see Fig. 5) and mostly those that caused serious material
or human damage. On the other hand, at present, the occurrence of strong winds of the 1st category
(fresh and strong breezes) in all meteorological stations in Poland is about ten times more frequent
than the sum of categories 2 and 3 (see Fig. S5). For this reason, time analysis of occurrences of most
extreme wind categories seems most reliable, not only for the study of historical periods but also when
we compare their frequencies in historical and contemporary periods. It is also not possible, based on
the documentary evidence, to reconstruct strong winds for the individual places in Poland as it is
normally done in the case of systematic instrumental measurements. Therefore, all series of strong
wind frequencies are presented here for the entire area of Poland.
The lower gust wind speed threshold, which was assumed for central Europe as potentially
dangerous for the destruction of buildings, forests, gardens, etc., is $17\,ms^{-1}$ (Brázdil and Dobrovolny
2001; Lorenc 2012). Therefore, it is possible to approximately compare the frequency of strong winds
above this threshold obtained from contemporary instrumental observations with the summed
frequency of historical winds assigned to categories 2–5. In the case of category 4 (squall), only those
cases for which damage was noted (8 cases) were taken. The results of the comparative analyses
presented below must, however, be limited only to similarities and differences in the annual cycle and
spatial distribution in the study area. A reliable comparison of absolute values of the frequency of
strong winds is impossible due to the undetectability of probably a significant number of events in the
study's historical period. This will only be possible for some isolated periods for which daily weather
records are available. For Poland, such series exist for selected periods in the 17th and 18th centuries
(for details see Introduction) and will be the subject of our analysis in a separate article.
In Poland, the most frequent winds are from the western sector (from SW to NW), in particular
from the NW direction, while the least frequent winds are from the sector from N to ESE (see Fig. 15.4
in Wibig 2021). According to investigations of Lorenc (2012), about 80% of strong winds $\geq17$ ms$^{-1}$ in
Poland in the period 1991–2005 were associated with the occurrence of NWc circulation type (i.e.,
wind blowing from NW direction within cyclonic pattern) according to Lityński's (1969) classification.
Also, the hurricane winds (>33 ms$^{-1}$) in Poland in the period 1971–2005 were mostly associated with
NWc type and then with Wc type. As a result, both the greatest measured winds and the greatest
frequency of them are characteristic for north-western, western and south-western Poland, in
particular in the cold half-year (Lorenc 2012; Wibig 2021, see also Figs 10–12). Such spatial distribution
of strong winds was also found for the study's historical period when the greatest frequencies were
noted in the Baltic Coast and Pomerania and Silesia regions (see Fig. 4). In the Greater Poland region



(central western Poland) this was not found due to a very small number of historical sources available
(see Fig. 2). It is worth also noting some similarity in annual cycles of strong wind occurrences in the
historical and modern periods. In both periods, the greatest frequencies occurred in the cold half-year,
and the lowest in summer. However, in the historical time, a greater frequency was observed in
autumn than in winter, i.e. opposite to the modern time (cf. Figs 4 and 11). This can be related to
greater continentality of climate in medieval times than at present (see Sadowski 1991; Przybylak
2016; Przybylak et al. 2023). The winters were clearly colder than today (see Przybylak et al. 2005,
2023) and were connected with negative values of the NAO (Przybylak et al. 2003). Such a circulation
pattern in the Atlantic-European sector allowed more frequent advection of cold air masses to Poland
from eastern and northern sectors within anticyclones coming from eastern Asia (Siberia) and the
Arctic. On the other hand, positive NAO conditions often bring high storminess in Europe (Ommel 2015
and references therein).

From the mid-15th century, a decadal number of sources and, as a consequence, the number of

occurrences of strong winds seems comparable with those found in the 16th century (see Fig. 3).
Therefore, for this period, it is possible to make a comparison with analogical investigations made for
other parts of Europe. Unfortunately, the comparison is limited to only a few existing works presenting
results, in particular for the 15th century (see Introduction). For Czech Lands, Brázdil et al. (2004) found
only 24 cases of strong winds for this century, i.e. 40% fewer than we found for Poland (40). But the
main difference is that, in the Czech Lands, the strong winds were connected with thunderstorms
(convective storms), while in Poland, this category was noted only five times. A better and more
reliable comparison is possible with a number of storm events occurring in the coastal area of Belgium
and the south-west Netherlands provided by DeKraker (2013) using town accounts informing about
repairing the damage of dikes, piers, groynes, and quay walls after storms. In the second half of the
15th century, in both areas (Poland and western Europe), the greatest numbers of strong winds were
noted in the first and the last decades (cf. Graph 1 in DeKraker 2013 and Fig. 3 in this study). Also, a
good agreement is seen in the 16th century. Both in Poland and coastal areas of Belgium and the
Netherlands, strong winds were more common in the second half of the century than in the first.
Results presented for the Czech Republic (Brázdil et al. 2004) and Germany (Glaser 2013) also confirm
this finding. The greatest difference is noted for the second decade of the 16th century, which was
exceptionally rich in storm events in Belgium and the Netherlands in comparison to neighbouring
decades. In Poland, the maximum in this decade is also seen, but it is not as great as in the case of
Belgium and the Netherlands. On the other hand, in the Czech Republic, this maximum is not seen. In
Germany, there was a steady increase in the number of strong winds until about 1575 and then a
decrease by the end of the century. Such a tendency in the occurrence of strong winds in the last three



decades of the 16th century was also noted in Poland (see Fig. 3). In both countries, the maximum of
strong winds in the 16th century occurred in the 1570s.
Comparison of potential periods with strong winds (usually reconstructions of high/low storminess
periods) with results obtained using other different natural proxies (e.g., sand dune development,
windblown sand in peat bogs or marshes, etc.) allows only for a very rough comparison. For example,
the reconstruction of the start of the great storminess period in the mid-16th century (also seen in the
documentary evidence from Poland, Czech Republic, Germany, and Belgium and the Netherlands) was
found for Scandinavia (De Jong et al. 2006; Clemmensen et al. 2008), the northwest Mediterranean
(Sabatier et al. 2012) and Portugal (Costas et al. 2012).

5. Conclusions and final remarks
There are quite a few mentions of strong winds in historical periods, but as can be seen from the
presented literature review, this important element has been studied in a long-term perspective to
only a small extent. One of the most important reasons for this is certainly the great dynamics over
time and spatial variability of the occurrence of strong winds in Europe, including Poland. This last
feature is especially important in the case of convective storms, which are local in nature. The
mentioned features of the occurrence of strong winds, together with the significantly changing
number of available historical sources, especially before 1500, significantly hamper the proper
recognition of this important element of the climate based on documentary evidence. As a result,
reliable examination of changes in the occurrence of strong winds compared to the modern period is
extremely difficult and limited but still possible.
Taking into account the following reservations, we present below the most important research
results:
- A reliable estimate of some characteristics of the occurrence of strong winds in Poland seems
possible since the mid-15th century onwards,
- The highest number of strong winds occurred in the second half of the 15th century and particularly
in the second half of the 16th century. The decade with the most significant number of strong winds
was 1571–80 (14) (see Fig. 3),
- For each season, the greatest number of strong winds was found for the Baltic Coast and Pomerania
region, and then for Silesia and Lesser Poland (Fig. 4),
- Strong winds were noted most often in autumn and winter in Silesia and in the Baltic Coast and
Pomerania regions (two regions for which there is enough information to estimate the annual cycle),





and in summer in the Lesser Poland region (Fig. 4b). August (13 cases) and March (12) were most
abundant in strong wind occurrences in Poland, but the stormiest period of the year was from October
to March, with at least ten cases in each month except February (Fig. 4a),
- In the entire study period (1281–1600), and also in two subperiods (1281–1500 and 1501–1600), the
most frequent were storms and gales and the least frequent were fresh and strong breezes (Fig. 5, Fig
S1), which were most common in autumn and in winter,
- All categories of strong winds in the study period occurred with the greatest frequency in the Baltic
Coast and Pomerania region, except squalls, which were most frequently noted in the Silesia region
(Fig. 6). In the 16th century, however, squalls were most common in Lesser Poland (Fig. S2),
- Damage of category E2 (areally extended damage) was more than three times more frequent than
category E1 (damage of lesser extent). About one third of the notes mentioned strong wind occurrence
and did not contain information about the damage. The greatest and most frequent damage was noted
in autumn (37.2%) and the smallest in spring (19.0%) (Fig. 7, Fig. S3),
- Out of all distinguished categories of damage, DB (26.0%) and DL (10.5%) were most frequent, while
the least frequent was DO (1.7%) (Fig. 9). About 60% of all damages mentioned in the weather notes
were found for the cold half-year, but particularly for autumn (39.2%). The analysis reveals that the
most frequently noted categories of damage in Poland (DB, DL) were also noted in all analysed regions,
with a maximum in Silesia and Baltic Coast and Pomerania regions.
- Spatial distribution of strong winds estimated for the studied historical period based on documentary
evidence, i.e. the greatest frequencies of strong wind occurrence in the Baltic Coast and Pomerania
and Silesia regions (see Fig. 4) as well as the annual cycle (Fig. 5) are similar as today (Lorenc 2012;
Wibig 2021; see also Figs 10–12). These facts seem to confirm, to some extent, the reliability of the
results obtained for the historical period,
- The time changes in occurrences of strong winds in Poland in the period since the mid-15th century
were found to correspond well with other European countries (e.g., Czech Republic, Germany, and the
Netherlands and Belgium) using documentary evidence. Other proxy data used to reconstruct
storminess changes in some parts of Europe in historical times (e.g., in Scandinavia, Portugal, and the
south-west Mediterranean) also shows a good correspondence with the presented results for central
and western Europe,
- The "fresh and strong breeze" category of strong winds in historical times was rarely noted by
chroniclers, because winds of this category did not cause any material or human damage. On the other
hand, this category of strong winds delimited using measurement data is, at present, about ten times





more frequent than gales and storms taken together. Therefore, this category of strong winds
recorded sporadically during the historical period cannot be reliably compared to current conditions.
This will probably only be possible for isolated periods for which daily weather observations are
available.

The creation of databases on extreme phenomena and events in historical periods, as in this
case of strong winds, is extremely important in order to be able to recognise the natural range of their
temporal changes and to learn about spatial variability and their changes over time. The high
variability, both in time and space, of the extreme phenomenon under study requires that significant
temporal and spatial coverage be obtained for the correct and reliable identification of its features in
Europe from a long-term perspective. The review of the state of knowledge on this subject made in
this article clearly proves that it is very limited and, therefore, urgently requires intensified work,
especially using documentary evidence, which is undoubtedly the most accurate source of information
for the last millennium.

The same method of processing data on strong winds available in the documentary evidence
of individual European and other non-European countries would significantly facilitate the recognition
of this extreme phenomenon over a larger area and the comparison of results. For these reasons, in
this article, we applied the proposal given in the work of Brázdil et al. (2004), which we only slightly
modified to reflect the Baltic Sea's influence on the coastal part of Poland.

**Competing interests.** The authors declare that they have no known competing financial interests or
personal relationships that could have appeared to influence the work reported in this paper.

**Acknowledgements.** The research work of RP, JF, PO and PW was supported by grant funded by the
National Science Centre, Poland No 2020/37/B/ST10/00710. The work of AA was supported by funds
from IDUB Research Group *Weather and Climate: Reconstructions and Future Scenarios*. We would
like to thank Dr Babak Ghazi for preparing Fig. 1.

**Author contributions. Rajmund Przybylak**: Conceptualisation, Methodology, Investigation, Data
collection and selection, Database construction, Formal analysis, Literature review, Interpretation of
results, Writing – original draft, Writing – review & editing, Funding acquisition, Project
administration. **Andrzej Araźny**: Conceptualisation, Methodology, Investigation, Data collection and
selection, Formal analysis, Validation, Visualisation, Interpretation of results. **Janusz Filipiak:**
Conceptualisation, Methodology, Investigation, Data collection and selection, Database construction,
Validation, Interpretation of results, Writing – original draft. **Piotr Oliński:** Conceptualisation,
Investigation, Data collection and selection, Database construction, Visualisation, Validation.
**Przemysław Wyszyński**: Software, Visualisation, Validation, Formal analysis, Data collection and
selection, Database construction. **Artur Szwaba:** Database construction.

**Financial support.** The work was supported by the National Science Centre, Poland, project No.
2020/37/B/ST10/00710.



**Data availability**. Datasets for this research were derived from the following public domain resources:

1. Repository for Open Data (RepOD), Nicolaus Copernicus University Centre for Climate Change Research collection, as cited in Przybylak et al. (2025), https://doi.org/10.18150/W6PMBQ

2. The Institute of Meteorology and Water Management (IMGW-PIB) website: https://danepubliczne.imgw.pl/

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
