# Peer review of "Strong wind occurrence in Poland from the 13th to 16th centuries based on documentary evidence"

_EGUsphere, 2025_

## Author Comment (AC1)

Dear Editor

We would like to thank you and the two anonymous reviewers for providing positive feedback and constructive comments about our manuscript. We believe that these comments helped us improve the description of our work.

All comments were carefully considered, and the detailed corrections/modifications are listed below, point by point. In addition, the manuscript was reviewed by a native speaker to meet the journal's required English standard.

Reviewer's comments (RC1) are in black, our answers in blue font.

RC1: 'Comment on egusphere-2025-1042', Anonymous Referee #1, 30 Apr 2025

I think this paper is needed and it is very important and a useful starting point for the scientific community. Wind is still relatively under-researched, and every field has to start somewhere. I believe this was a strong beginning for the topic.

Thank you very much.

It's a good idea overall.

Thank you very much.

There's no need to include the word enlightenment (line 99).

Was deleted.

I like the division into the three categories of wind and how they are described — I hope this classification can become generally accepted.

Thank you very much.

The database is very useful — great initiative!

Thank you very much.

Table 1 is a good example.

Thank you very much.

Figure 2 could benefit from a clearer caption. This applies to several figures — more self-explanatory captions would improve them (e.g., Figures 3, 12, and 13).

Thank you for your suggestion.

Changes introduced are the following:

Fig. 2 – We revised the caption to:

Fig. 2. Number (n) of weather notes for Poland for seasons (a) and regions (b), 1281–1600. Explanation: "N/A season" – specific season is not available (unknown); the category "Poland" encompasses all notes for which the region is unknown

We trust that this addresses the issues the reviewer was picking up on.

The caption for Figure 3 has been revised to:

Fig. 3. Decadal (a) and 50-year (b) number of occurrences [n] of strong winds of all categories in Poland, 1281–1600.

And similar other figure captions have been revised in line with Figure 3.

We modified captions in Figures 11-13.

Fig. 11. Year-to-year course of the annual number [n] of gust winds of >17 ms$^{-1}$ at selected stations representing six studied regions in Poland, 1993–2022

Fig. 12. Annual course based on the average monthly number [n] of gust winds of >17 ms$^{-1}$ at selected stations representing six studied regions in Poland, 1993–2022

Fig. 13. Maximum monthly wind gust speeds (ms$^{-1}$) at selected stations representing six studied regions in Poland, 1993–2022

The comparison with the contemporary period is a bit poorly explained and could be clarified.

Thank you for your suggestion. Some new additional information has been added.

Figure 4 is very good.

Thank you very much.

---

## Author Comment (AC2)

Dear Editor

We would like to thank you and the two anonymous reviewers for providing positive feedback and constructive comments about our manuscript. We believe that these comments helped us improve the description of our work.

All comments were carefully considered, and the detailed corrections/modifications are listed below, point by point. In addition, the manuscript was reviewed by a native speaker to meet the journal's required English standard.

Reviewer's comments (RC2) are in black, our answers in blue font.

RC2: 'Comment on egusphere-2025-1042', Anonymous Referee #2, 26 May 2025

This is a strong paper that makes an important contribution through its presentation and analysis of new research into historical documentary evidence for the occurrence of strong winds in Poland between the 13th and 16th centuries. The paper first presents the knowledge gap related to this kind of evidence and discusses the various issues related to the collation of textual records for the period as well as the methodology adopted by the authors to create a database of documentary records relating to strong wind occurrence. Next the paper presents the results of this work, dividing the results in terms of the spatial occurrence of strong winds and the magnitude and impact as reported in the textual sources (where possible). Results of modern instrumental data recording strong wind occurrence are also presented and discussed for comparison. The discussion and conclusion usefully bring this data together and importantly highlight that the data gathered from historical sources both matches the general pattern seen in modern instrumental data as well as agreeing with data gathered from textual sources from neighbouring European countries which experience weather patterns driven by similar systems. The data reported in this paper, therefore, seems to reflect genuine patterns which can usefully be used to understand strong wind occurrence over the timeframe of the study. The methods and analysis are convincingly presented throughout and take into account past work on the topic in neighbouring countries while pointing to possible future avenues for research.

I do not have any significant suggestions in relation to the content, the paper is well structured and easy to follow.

Thank you.

A minor technical question I have relates to lines 73-75. Discussing the results of Outten and Sokolowski, the authors state that: "they found increases in the return period, i.e. more frequent extreme episodes projected for Northern, Central and Southern Europe throughout the 21st century." However, if the return period is increasing (the length of time between extreme episodes is getting longer?), does this not mean that extreme episodes would become less, not more, frequent?

Thank you for your remark. The text in our manuscript is correct as per the findings presented by Outten and Sokolowski (2021). They write: 'However, for more general assessments the picture is more clear with increases in the return period (i.e. more frequent) extreme episodes projected for Northern, Central and Southern Europe throughout the 21st century'. They clarify explicitly that, by the vague phrase 'increases in the return period', they mean not an increase in lengths between extreme episodes, but an increase in the frequency of extreme events. Accordingly, we have

removed our use of their ambiguous phrase and stuck with their clarifying phrase. '…they found that more frequent extreme episodes were projected for Northern, Central and Southern Europe throughout the 21st century.'

The English is almost all completely understandable but there are several errors that should be corrected prior to publication. I list a few below, but this is not an exhaustive list.

All indicated language points have been addressed, either as per the reviewer's recommendation or as per our specific note below. Furthermore, a full final language edit was conducted to repair other language errors and improve the informational flow and style of the article.

 Line 38: 'second' is in the wrong position, better phrasing would be "they are the second most dangerous natural phenomenon after floods"

Done

Line 84: insert a 'the' before 1950s e.g. "...they all cover some periods since the 1950s and are written mainly in Polish and therefore..."

Done

Line 131-132: The 'and' between 'north and the Sudetes' should be replaced with a 'to' e.g. "Poland is a Central European country stretching from the Baltic Sea in the north to the Sudetes and Carpathian Mountains in the south (Fig. 1)"

Done

Line 132-133: winds should be singular and the 'the' before analysis is unnecessary e.g. "To more precisely estimate the spatial changes in strong wind occurrence, analysis was also made for six historical-geographical regions: Baltic Coast and Pomerania, Masuria and Podlasie, Greater Poland, Masovia, Silesia, and Lesser Poland (Fig. 1)."

Done

Line 135-136: A 'the' should be inserted before 'contemporary' e.g. "Data from the contemporary period representing all the mentioned regions were collected for 12 meteorological stations – two for each region."

Done

Line 183-184: 'only one' not 'one only' e.g. "A similar situation is noted for both sub-periods with only one exception being the category E1 in the period 1281–1500 (Fig. S4)."

Changed to 'the only'

Line 200-202: damages is singular e.g. "About 60% of all damage mentioned in the weather notes were found for the cold half-year, but particularly for autumn (39.2%)."

We don't want to imply damage in the uncountable sense, as we are not assessing here the amount of damage but the number of reports. We have revised to 'mentions of damage'.

Line 224-225: This sentence is difficult to follow and should be rephrased e.g. "Also, helpful knowledge can be observed about the values of the greatest speed of strong winds in Poland in modern times."

Revised to 'It will also be helpful to know about the greatest speeds of strong winds currently being observed in Poland.'

Overall, this is a useful paper which presents new data that is important for understanding the past occurrence of strong winds in Poland. It should be published once these very minor issues are addressed.

Thank you.